# A Novel Color Image Encryption Algorithm Based on Hyperchaotic Maps and Mitochondrial DNA Sequences

**DOI:** 10.3390/e22020158

**Published:** 2020-01-29

**Authors:** Heba G. Mohamed, Dalia H. ElKamchouchi, Karim H. Moussa

**Affiliations:** 1Electrical Department, College of Engineering, Princess Nourah Bint Abdulrahman University, Riyadh 11671, Saudi Arabia; 2Electrical Department, College of Engineering, Alexandria Higher Institute of Engineering and Technology, Alexandria 21421, Egypt; 3Information Technology, College of Computer and Information Sciences, Princess Nourah Bint Abdulrahman University, Riyadh 11671, Saudi Arabia; 4Electrical Department, College of Engineering, Horus University Egypt, New Damietta 34518, Egypt; Khassan@horus.edu.eg

**Keywords:** hybrid chaotic, image, encryption, decryption, secured communications, DNA, mitochondrial genome

## Abstract

Multimedia encryption innovation is one of the primary ways of securely and privately guaranteeing the security of media transmission. There are many advantages when utilizing the attributes of chaos, for example, arbitrariness, consistency, ergodicity, and initial condition affectability, for any covert multimedia transmission. Additionally, many more benefits can be introduced with the exceptional space compliance, unique information, and processing capability of real mitochondrial deoxyribonucleic acid (mtDNA). In this article, color image encryption employs a confusion process based on a hybrid chaotic map, first to split each channel of color images into n-clusters; then to create global shuffling over the whole image; and finally, to apply intrapixel shuffling in each cluster, which results in very disordered pixels in the encrypted image. Then, it utilizes the rationale of human mitochondrial genome mtDNA to diffuse the previously confused pixel values. Hypothetical examination and trial results demonstrate that the anticipated scheme exhibits outstanding encryption, as well as successfully opposes chosen/known plain text, statistical, and differential attacks.

## 1. Introduction

Owing to vast development in the modern digital era, the world has grown to be a worldwide community sharing private information. Because of this, people use images instead of long texts for sharing this kind of information securely and privately [1]. Therefore, the protection of such images has become one of the vital concerns in medical, military, and many other areas. Recently, many cryptographic algorithms depending on chaotic systems have been used in image encryption [2]. One of the recognized methods employed before image transmission is to convert an image into an unintelligible form, so that the transmitted message of the distorted image will not be discovered if it has reached an unsought end [3,4]. The encryption process is done with the help of a key, which produces variation on the pixel values of the image and destroys the apparent image attribute. Another encryption process is the scrambling image technique. In this technique, all pixel values are repositioned between different pixels of the image to be different from the original plain image, for example, Arnold transform, exponential transformation, geometry transform, and so on. These transformations have definite regularity, although the results of pixel distribution are different after scrambling, due to changes in only the position of pixels, without changing gray values. Arnold’s map has the property of periodicity. After running the algorithm for a certain number of times, the original image can be restored. Therefore, it has a weakness for the system [5]. However, whilst [6] demonstrated the powerful recovery of an image encryption system based on an Arnold map and Lü map, their encryption system was susceptible to several attacks, such as plaintext and chosen-text attacks when it depended on only the permutation of pixel values [7]. As a result, these attack pixel values have to be rearranged and vary their position. Rearrangement of the values of pixels is called the confusion process, while changing the pixel values is known as the diffusion process. Chaos is a multi-disciplinary hypothesis which states that during the randomness of the chaotic complex technique, the chaotic system generates sequences due to its initial parameters’ sensitivity, periodic and pseudo randomness, and long-term volatility [8], which encouraged us to employ the chaos theory in our new proposed scheme.

In recent years, several scientific papers have concentrated on researching chaotic image encryption. In 2015 [9], S. S. Askar, A. A. Karawia et al. presented an image encryption system based on a chaotic economic map, and the simulation outputs a display so that the proposed algorithm can utilize the same secret keys to encrypt and decrypt the images successfully. The security analysis shows that the encrypted images have a good performance, but cannot resist noise attacks. After one year, Abbas proposed a new image encryption technique based on unconventional element analysis and Arnold’s Cat Map [10], which was easily achieved and afforded an efficient and secure approach for image encryption. However, the algorithm depends on the encryption of square images only, so the domain of application is remarkably restricted. Later, in 2017, Shaikh, Chapaneri et al. [11] proposed a color image encryption algorithm, which is a single round-based hyperchaotic system due to bi-directional pixel diffusion that contributes towards an increased security and improved efficiency. Recently, in 2019, Chenghai Li et al. used the “transforming-scrambling-diffusion” model to introduce a hyperchaotic color image encryption algorithm that depends on converting the pixel values into gray codes before scrambling [12]. In the same year, Priya Ramasamy et al. suggested an improved logistical map while using a chaotic maps function and uncomplicated encryption techniques, such as block scrambling and modified zigzag transformation, for encryption phases [13]. Recently, deoxyribonucleic acid (DNA) was applied to an image chaotic cryptosystem [14,15,16,17,18,19,20,21,22], as a result of its numerous superior characteristics, such as its enormous parallelism, vast storage, small power consumption, etc. In 2016, Kulsoom et al. extracted the most significant bits and the least significant bits for each pixel of an image and performed DNA computing for them. Kulsoom et al.’s algorithm has a low robustness against noise because most digits of each pixel are not changed [23]. After that, in 2018, Ran Wei et al. proposed an image cryptosystem that depends on the combination of DNA encoding and several chaotic maps [24]. The performance analysis illustrated that the system could greatly enhance the sensitivity of plain images and secret keys, but the algorithm was complex and the necessities for hardware were comparatively high. In 2020, J. Wu, J. Shi, T. Li, proposed a novel image encryption algorithm based on a hyperchaotic system and variable kernels for the confusion stage and a DNA technique for the diffusion stage [25].

Mitochondrial DNA (mtDNA) is a small part of the DNA of organelle cells within eukaryotic cells [26]. Eukaryotic cells are organisms that have a nucleus enclosed within membranes that are used to transform chemical energy into a formula that cells can use [27]. In humans, there are 16,569 base duos of mitochondrial DNA to code only 37 genes. The first major part of the human genome that can be sequenced is human mitochondrial DNA [28]. mtDNA is congenital and uniquely from the mother in several organisms, including humans. Since animal mtDNA develops more rapidly than nuclear genetic symbols [29,30], it signifies a mainstay of phylogenetic and evolutionary biology. mtDNA allows inspection of the kinship of inhabitants. Therefore, it has become vital in molecular biology and biotechnology.

In this article, a new image cryptosystem is presented to encrypt a color image using identical encryption and decryption schemes. Both of these processes consist of confusion stages employing a hyperchaotic system, diffusion stage, and mtDNA. The proposed encryption algorithm was applied on three channels of a color plain image to increase uncertainty in the plain image. From the numerical analysis, the proposed algorithm was seen to be robust against various attacks, for example, chosen/known plain text attacks, brute force attacks, differential cipher image attacks, and entropy attacks. It has a large key space and is sensitive to minimal change in the chosen secret key. The encryption scheme was also compared with newly developed encryption techniques. The rest of the article is organized as follows. Section 2 details a brief description of the hyperchaotic system, mtDNA, and DNA encoding. The explanation and argument of the proposed encryption/decryption algorithm are introduced in Section 3. Section 4 gives the numerical simulation results, while the security act of the proposed scheme is analyzed in Section 5. Finally, Section 6 concludes the article.

## 2. Preliminaries

### 2.1. Hyperchaotic System

A hyperchaos system, technologically, advanced from chaos. Its dynamical structure is more complicated compared with other chaotic systems. In addition, hyperchaos enhances a higher level of randomization and uncertainty. By adding two or more positive Lyapunov exponents, the hyperchaotic system can be distinguished from chaos. Hyperchaos exists in four-dimension nonlinear systems. Owing to its simpler formula and higher effectiveness, the chaotic system has a smaller key space and lower complexity. Consequently, the chaotic system has lower security protection. On the contrary, the hyperchaotic system has more state variables, a larger key space, and complex nonlinear behavior, resulting in higher security protection. The hyperchaotic system can be determined as follows [31]:(1)$$\begin{array}{l} \overset{\cdot }{x_{1}}=\alpha (x_{2}-x_{1})+\lambda_{1}x_{4},\\ \overset{\cdot }{x_{2}}=\xi x_{1}-x_{1}x_{3}+\lambda_{2}x_{4},\\ \overset{\cdot }{x_{3}}=-\beta x_{3}+x_{1}x_{2}+\lambda_{3}x_{4},\\ \overset{\cdot }{x_{4}}=-\tau x_{1}, \end{array}$$
where $\alpha ,\beta ,\xi ,\tau ,\lambda_{1},\lambda_{2}$, and $\lambda_{3}$ are the control parameters of the 4-D hyperchaotic system. The system presents hyperchaotic behavior when the control parameters are $\alpha =35,\beta =3,\xi =35,\tau =5,\lambda_{1}=1,\lambda_{2}=0.2$, and $\lambda_{3}=0.3$

### 2.2. Fundamentals of Mitochondrial DNA

Mitochondrial DNA is a type of DNA and is a small chromosome in a circular form found inside organelles known as mitochondria. Mitochondria are found in cells and have been determined in all unpredictable or eukaryotic cells, as well as plants, creatures, growths, and single-celled protists, which include their own mtDNA genome. In several creatures, mtDNA is a two-fold stranded particle that is shaped around the genome. Every mitochondrion can have several duplicates of the mtDNA genome. In human embryonic advancement, the variety of mitochondria and the substance of mtDNA in every mitochondrion have an impact on the creation of oocytes, preparation of the oocytes, and early embryonic development and enchantment [32].

For example, a simple sequence for the human mitochondrial genome is as follows:“GATCACAGGTCTATCACCCTATTAACCACTCACGGGAGCTCTCCATGCATTTGGTATTTTCGTCTGGGGGGTGTGCACGCGATAGCATTGCGAGACGCTGGAGCCGGAGCACCCTATGTCGCAGTATCTGTCTTTGATTCCTGCCTCATTCTATTATTTATCGCACCTACGTTCAATATTACAGGCGAACATACCTACTAAAGT…”

### 2.3. DNA Encoding

DNA encoding has a huge data capacity, so it can be used in cryptography [33]. In the transmission process, DNA can also be used to store data in image encryption. DNA computing depends on the DNA logic word, and only two digits are used to create four nucleic acid bases. Therefore, the information is stored in the form of these bases, which are Adenine (A), Cytosine (C), Guanine (G), and Thymine (T) [34]. In DNA cryptography, the four bases are used to capture the information. “A” and “T” are paired duos and the same is true for “C” and “G”. Table 1 displays the rubrics for DNA encoding. 

## 3. Proposed Cryptosystem

The structure of the proposed cryptosystem is demonstrated in Figure 1, and involves two main phases. In the first phase, the confusion phase, the image pixels are shuffled based on the generated hyperchaotic sequence discussed in the first part of Section 2. For more unpredictability and to increase the efficiency of encryption, the pixels’ positions are scrambled over the entire image, without changing the significance of the pixels, and the image becomes unknown. Hence, the primary and control factors of utilized chaotic maps serve as the employed undisclosed key. To improve the security further, the second stage, the diffusion stage, aims to diffuse the images’ pixel values with the decimal converted values of the mtDNA sequence using the Exclusive OR (XOR) operation.

### 3.1. Hyperchaotic Sequence Generation

We can use the hyperchaotic system discussed in Section 2.1 to generate the pseudorandom sequence. This is necessary for the field of cryptography, as both non-linearity and random performances make chaotic systems able to generate pseudorandom sequences. The following steps describe the sequence generation:To reduce the cross effects and raise the security, pre-iterate the hyperchaotic system $N_{0}$ times;The system is iterated another *M* × *N* times after $N_{0}$ iteration times. In each iteration j, four state values $\left\{x_{1}^{j},x_{2}^{j},x_{3}^{j},x_{4}^{j}\right\}$ are sorted, where j denotes the iteration index;During each iteration, each state value of the four-state values $\left\{x_{1}^{j},x_{2}^{j},x_{3}^{j},x_{4}^{j}\right\}$ is used to generate two different key values (${\left(v_{i}^{1}\right)}^{j}\in [0,255],$
i=1,2,3,4 and ${\left(v_{i}^{2}\right)}^{j}\in [0,255],$), which are calculated by
(2)$${\left(v_{i}^{1}\right)}^{j}=\mathrm{mod}\{floor([(|x_{i}^{j}|-floor(|x_{i}^{j}|))\times 10^{15}]/10^{8}),256\},\;\;\;\;i=1,\ldots ,4,$$
(3)$${\left(v_{i}^{2}\right)}^{j}=\mathrm{mod}(floor(\mathrm{mod}\{[(|x_{i}^{j}|-floor(|x_{i}^{j}|))\times 10^{15}]/10^{8}\}),256),\;\;i=1,\ldots ,4,$$
where mod (⋅) indicates the modulo process; Concatenate Equation (2) with Equation (3) to get $v{\;}^{j}$:(4)$$v^{j}=[{\left(v_{1}^{1}\right)}^{j},{\left(v_{2}^{1}\right)}^{j},{\left(v_{3}^{1}\right)}^{j},{\left(v_{4}^{1}\right)}^{j},{\left(v_{1}^{2}\right)}^{j},{\left(v_{2}^{2}\right)}^{j},{\left(v_{3}^{2}\right)}^{j},{\left(v_{4}^{2}\right)}^{j}];$$These sequences are concatenated after the whole iteration with Equation (5) to obtain *K*: (5)$$K=[v^{1},v^{2},\ldots \ldots ,v^{M\times N}].$$

One component in *K* can be represented by $K_{i},i\in [1,8MN].$

### 3.2. Image Encryption

Let M×N×3 represent the whole dimension of the contribution color image **P**, as displayed in Figure 2a;Obtain the three channels R, G, and B from the input image;Divide each channel for both the row and column into p×q clusters, where p=2,4,8,16 and q=2,4,8,16 randomly depend on the generated chaotic sequence. For example, when p=2 and q=4 the resulting image is split into eight clusters, as shown in Figure 2b;Shuffle all the clusters using the generated chaotic sequence *K,* as illustrated in Figure 2c;Apply a chaotic sequence for each cluster to shuffle the pixels within every cluster, as clarified in Figure 2d;Concatenate these permuted clusters to formulate the shuffled image with the same size of the novel colored image illustrated in Figure 2e;To get new values of pixels, apply the diffusion process as follows:
Read the first 64 × 64 × 4 mitochondrial DNA sequence “gatcacaggtctatcaccctattaaccactca……”;Convert it to a binary sequence using the rule discussed in Table 1:“{0001101101110100001011100110110111111110011010010111110……}”;Divide the sequence into 8-bit slices: {00011011 01110100 00101110 01101101 11111110…};Convert each slice into its decimal representation: [27 116 46 109 254 105 125…];Formulate a 64 × 64 rdDNAstring matrix with the above decimal sequences;Generate a 3 × 64 × 64 matrix for the three channels R, G, and B, as in
(6)DMat(:,:,i)=mod[i×mod[sum(sum{sum(Image)}),256]×rdDNAstring,256]  for i=1,2,3;Generate 128 × 128 × 3 using
(7)lDMat(:,:,i) = [DMat(:,:,i) DMat(:,:,i)′;DMat(:,:,i)′ DMat(:,:,i)′]  for i=1,2,3,
where [ ]’ is the transpose of the matrix;Generate 256 × 256 × 3 using
(8)llDMat(:,:,i) = [lDMat(:,:,i) lDMat(:,:,i)′;lDMat(:,:,i)′ lDMat(:,:,i)′]  for i=1,2,3;Resize the generated matrix to the same size of the concatenated image; Apply the XOR operation between the concatenated shuffled pixel values and the decimal converted values of mtDNA classification r1 and r2 to get the diffused encrypted image, as shown in Figure 2f.

## 4. Simulation Results

Experimentation was carried out on Lena and baboon images. The Matlab (R2015a) software (MathWorks, Natick, MA, USA) was used to apply the proposed algorithm. In the hyperchaotic map, the control parameters $\alpha =35,\beta =3,\xi =35,\tau =5,\lambda_{1}=1,\lambda_{2}=0.2$, and $\lambda_{3}=0.3$ were used to produce a chaotic sequence. The hyperchaotic sequences were arranged and the position of arranged sequences was used to split the plain image into n-clusters, shuffle the whole clusters, and scramble the pixels of each cluster. The utilized colored plain images were Lena, with dimensions of 256 *×* 256 pixels, as shown in Figure 3a, and Baboon, with dimensions of 512 × 512 pixels, as shown in Figure 3d. The corresponding cipher image was generated using the encryption algorithm and is shown in Figure 3b,e. Then, we decrypted the cipher image, as presented in Figure 3b,e, with correct undisclosed key ***K*** to get the perfectly reconstructed image shown in Figure 3c,f. 

## 5. Performance and Security Analysis

### 5.1. Key Space

The key space of our new encryption algorithm consisted of the initial conditions of hyper-chaotic systems $x_{1},x_{2},x_{3},x_{4},r_{1}$, and $r_{2}$. The value ranges were $x_{1}\in (-40,40)$, $x_{2}\in (-40,40)$, and $x_{3}\in (1,80)$, each with a step size of $10^{-13}$, while the value range was $x_{4}\in (-250,250)$, with a step size of $10^{-12}$. $r_{1}$ and $r_{2}$ are two 8-bit random numbers whose value range is [0, 255] with a single step size. These factors were used as undisclosed keys for both the encryption and decryption process. Therefore, the key space of the proposed algorithm is 1.6777 × $10^{64}$. The key space must be very large to restrict brute-force attacks; hence, it would take 2.03451 × $10^{52}$ days to crack the system. Therefore, the proposed cryptosystem is able to protect against brute-force attacks. 

### 5.2. Histogram Analysis

Histogram analysis is an important statistical feature for estimating the enactment of an image encryption cryptosystem. The histograms for the three channels of original images of Lena and Baboon and their corresponding generated cipher images are presented in Figure 4. The original image’s histogram is non-uniform and the characteristic peak is clear, and most of the image data were achieved smoothly. On the other hand, the recovered image is a perfectly reconstructed form of the original image, and the cipher image is noise-like and not correlated with the original plain image. The recovered image is a perfectly reconstructed version of the original image. In the cipher image, the histogram is very stable and smooth, which proves that no statistical data from the original plain image was included. Therefore, the proposed system can counterattack statistical and cipher image attacks.

### 5.3. Correlation Factor Analysis

The correlation factor (CF) is an important metric for calculating two adjacent pixels in the three directions of horizontal, vertical, and diagonal. *N* pairs of neighboring pixels are selected, and x,y are two neighboring pixels. Then, the calculation formula of the correlation factor can be given by
(9)$$CF=\frac{\sum_{i=1}^{N}\left(x_{i}-\bar{x})(y_{i}-\bar{y}\right)}{\sqrt{\sum_{i=1}^{N}{\left(x_{i}-\bar{x}\right)}^{2}}\sqrt{\sum_{i=1}^{N}{\left(y_{i}-\bar{y}\right)}^{2}}}.$$

Now, let *N* = 2000. The results of the correlation of the original and cipher images were computed and are presented in Table 2. The low value of the correlation factor for neighboring pixels in the three directions indicates that the neighboring pixels are uncorrelated. Figure 5 and Figure 6 display the correlation distribution of each pair of horizontally neighboring pixels. The correlation factor of the original image is high and close to 1. In contrast, the correlation factor calculated for the cipher image is low and close to 0. Therefore, these results verify the strength of the proposed cryptosystem.

### 5.4. Key Sensitivity Analysis

We can observe in the proposed algorithm the sensitivity of the generated secret key $x_{1},x_{2},x_{3},x_{4},r_{1}$, and $r_{2}$ by modifying and changing one bit in it. The wrong key can be generated by changing any element of secret key $x_{1},x_{2},x_{3}$ by $10^{-13}$ or $x_{4}$ by $10^{-12}$, or any value of $r_{1}$ and $r_{2}$ by 1. This small change leads to a totally different ciphered image. Therefore, the ciphered image will be wholly dissimilar from the original input image. In the proposed cryptosystem, we encrypted the original plain images twice to exhibit the sensitivity of the generated secret key. In the first encryption process, the plain images were encrypted with the original secret key, while in the second encryption process, the plain images were encrypted with the wrong key. The wrong key was generated from the original key. We ran the algorithm on the images of Lena and the baboon, and the effects are displayed in Figure 7 and Figure 8. As can be seen in the figures, the encrypted images with the wrong key are totally different from those encrypted with the exact key, which proves that the encryption algorithm has a high sensitivity to secret keys. 

### 5.5. Differential Attack

An opponent can get valuable information by altering several pixels of the original plain image. The number of changing pixel rate (NPCR) and the unified averaged changed intensity (UACI) are usually employed to evaluate the resistance of the encrypted plain image against differential raids. We utilized the presented encryption scheme to encrypt $P_{1}$ and $P_{2}$ to obtain their corresponding encrypted images, indicated by $C_{1}$ and $C_{2}$, with a unify secret key, where $P_{2}$ is the change in one pixel of the original plain image $P_{1}$. Then, the NPCR and UACI could be given by
(10)$$\mathrm{NPCR}=\frac{1}{W}\sum_{i=1}^{M}\sum_{j=1}^{N}\left|Sign(C_{1}(i,j)-C_{2}(i,j))\right|\times 100\%,$$
(11)$$\mathrm{UACI}=\frac{1}{W}\sum_{i=1}^{M}\sum_{j=1}^{N}\frac{\left|Sign(C_{1}(i,j)-C_{2}(i,j))\right|}{256}\times 100\%.$$

We chose randomly, changed the value of one pixel of the original input images 10 times, and calculated the NPCR and UACI of Lena and the baboon, respectively. The outputs are summarized in Table 3 and Table 4, which clarify that our new cryptosystem is very sensitive to variation arising in only one pixel value of the original image. In [36,37,38], NPCR and UACI are close to theoretical values, but they have a reduced noise attack performance. Compared with the proposed cryptosystem, it is clear that the proposed system achieves a high performance by exhibiting average performances very close to their notional amount.

### 5.6. Information Entropy

The information entropy, which reflects the insecurity of image data, was denoted by E(m) of matrix *m*, and evaluated by
(12)$$E(m)=-\sum_{i=0}^{255}P_{b}(m_{i})\log_{2}(P_{b}(m_{i})),$$
where $P_{b}(m_{i})$ denotes the probability of $m_{i}$. An accurate random image would yield 256-pixel values with an identical probability. Therefore, the theoretical value of the information entropy is closer to 8. Table 5 illustrates that the entropy values of cipher images of Lena and baboon are close to the theoretical value. Therefore, the outcomes prove that the proposed cryptosystem can efficiently counterattack the information entropy.

### 5.7. Avalanche Effect

When minor variation in the plain image or the secret key occurs, it will influence variation in the ciphered image. This influence is defined as the avalanche effect. To evaluate the avalanche effect of the proposed cryptosystem, the mean square error was evaluated for the two different cipher images $C_{1}$ and $C_{2}$ obtained from a slight change in the secret keys. The MSE can be given by
(13)$$\mathrm{MSE}=\frac{1}{W}\sum_{i=1}^{M}\sum_{j=1}^{N}({\left(C_{1}(i,j)-C_{2}(i,j)\right)}^{2},$$
where W=M×N is the dimension of cipher images.

From Equation (13), the MSE of the proposed encryption algorithm can be evaluated for cipher images using the original secret key and cipher images obtained by changing each element in the original secret key, as cited in Table 6 and Table 7, for both the Lena and baboon images, respectively. Cipher $C_{1}$ is obtained by changing only key $x_{1}$ and maintaining the other keys’ values. Similarly, ciphers $C_{2}$, $C_{3}$, and $C_{4}$ are generated by changing $x_{2}$, $x_{3}$, and $x_{4}$, respectively. The large values of MSE in Table 6 and Table 7 indicate that the proposed cryptosystem has a strong avalanche effect.

## 6. Conclusions

This article has presented a novel encryption and decryption algorithm for the purpose of verifying picture transmission over information correspondence frameworks. Both of them are indistinguishable with hybrid chaotic confusion procedures and the mtDNA diffusion process that diminish the equipment usage intricacy and improve framework security. The exhibited algorithm was shown to be powerful against chosen/known plain text attacks. The numerical investigations demonstrated that the proposed cryptosystem has a very large key space for opposing brute-force attacks and scrambled pixels appropriated haphazardly through the figured picture. Likewise, the proposed framework is sensitive to insignificant changes in the covert encryption key and can oppose the known plaintext, chosen plaintext, differential figure picture, and entropy attacks.

Finally, in terms of future research, we suggest additional simulation analysis of the chaotic performance by using a cosine chaotic model and another confusion technique to generate a robust chaotic image encryption technique with the addition of digital signature technology, for the sake of achieving more secure authenticated data transmission.

## Figures and Tables

**Figure 1 entropy-22-00158-f001:**
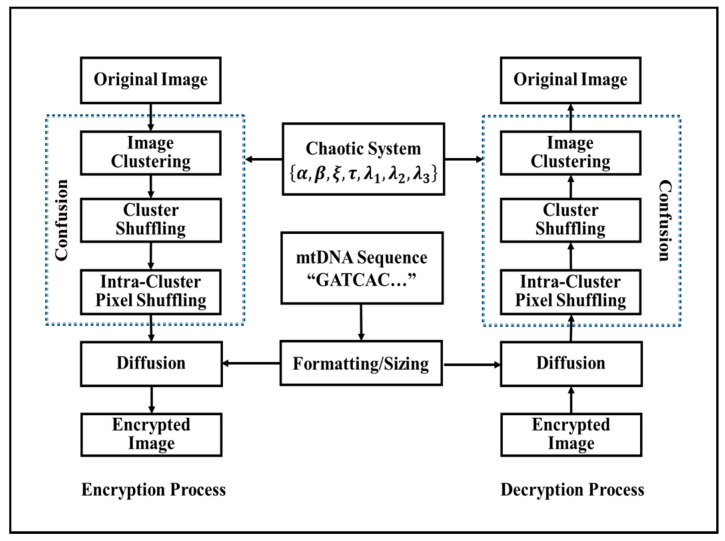
The proposed algorithm.

**Figure 2 entropy-22-00158-f002:**
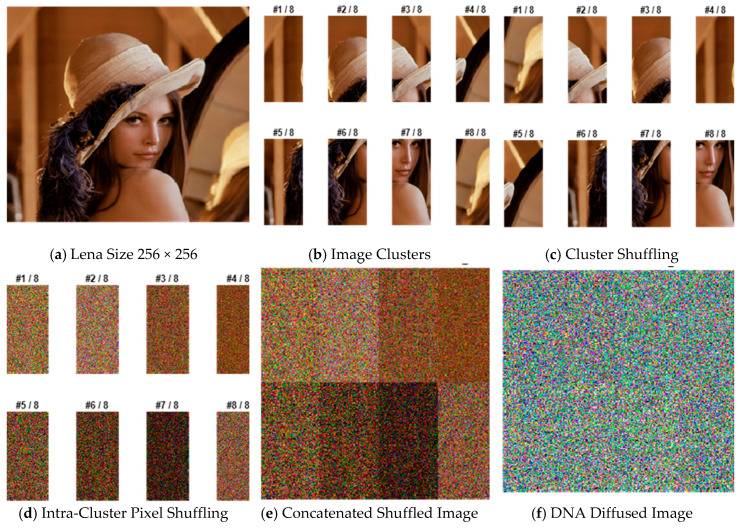
Simulation results of a Lena image.

**Figure 3 entropy-22-00158-f003:**
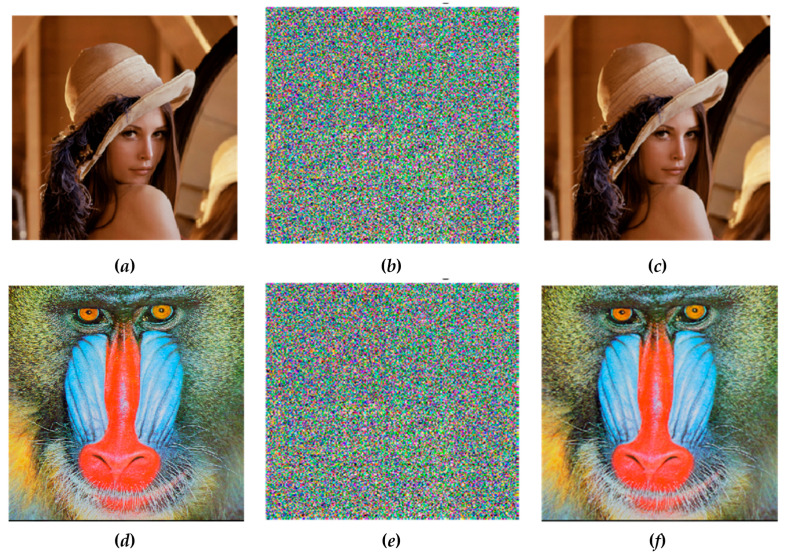
Simulation results. (**a**,**d**) Plain images; (**b**,**e**) generated cipher images of (**a**,**d**), respectively; and (**c**,**f**) recovered images.

**Figure 4 entropy-22-00158-f004:**
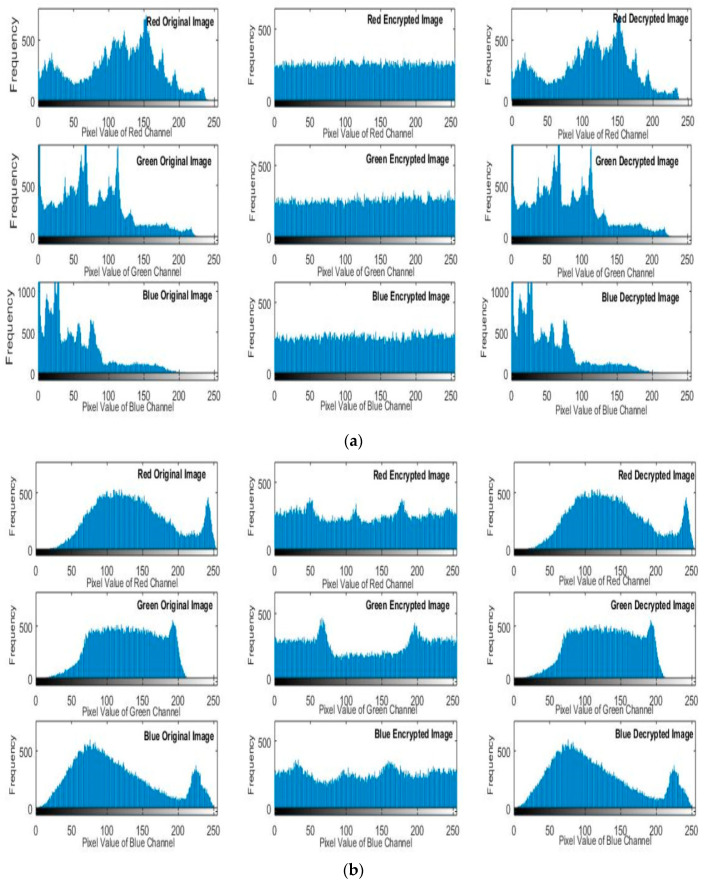
Histograms of colored plain images. (**a**) Histograms of Lena. (**b**) Histograms of baboon. The first column includes histograms of the original image, the second column includes histograms of the corresponding cipher image, and the third column includes histograms of the recovered colored image.

**Figure 5 entropy-22-00158-f005:**
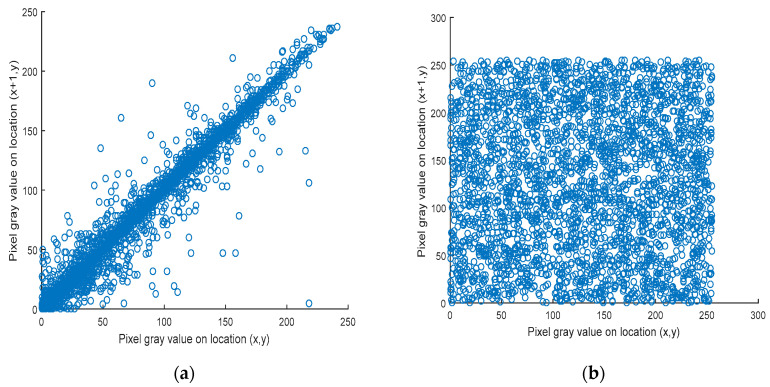
Simulation of the correlation factor. (**a**) Correlation in the horizontal direction for the Lena plain image; (**b**) correlation in the horizontal direction for the Lena cipher image.

**Figure 6 entropy-22-00158-f006:**
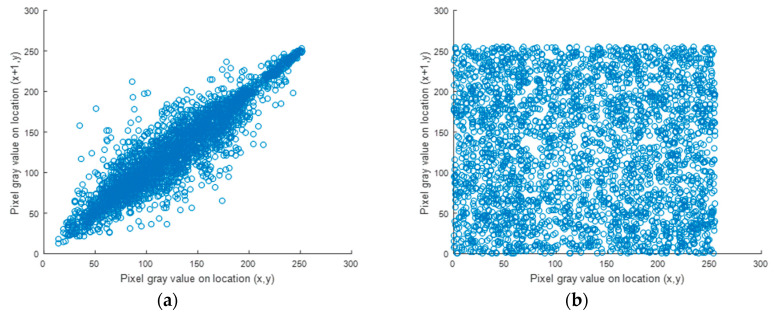
Simulation of the correlation factor. (**a**) Correlation in the horizontal direction for baboon plain images; (**b**) correlation in the horizontal direction for baboon cipher images.

**Figure 7 entropy-22-00158-f007:**
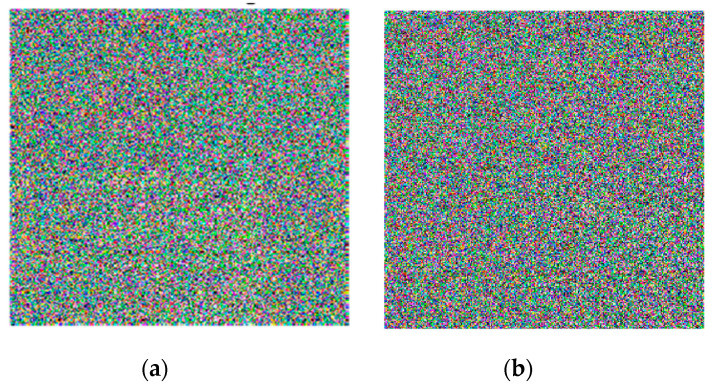
Key sensitivity analysis. (**a**) Encrypted Lena image using the exact secret key; (**b**) encrypted Lena image using the wrong key.

**Figure 8 entropy-22-00158-f008:**
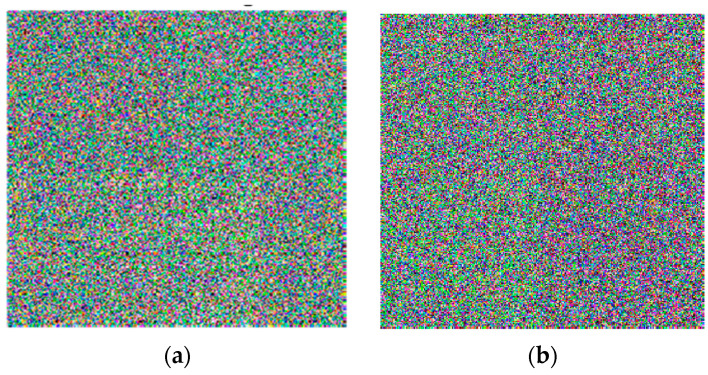
Key sensitivity analysis. (**a**) Encrypted baboon image using the exact secret key; (**b**) encrypted baboon image using the wrong key.

**Table 1 entropy-22-00158-t001:** Rubrics for DNA encoding [35].

Rule No.	DNA Nucleic Acids			
	A	T	G	C
Rb1	00	11	01	10
Rb2	00	11	10	01
Rb3	11	00	01	10
Rb4	11	00	10	01
Rb5	10	01	11	00
Rb6	01	10	11	00
Rb7	10	01	00	11
Rb8	01	10	00	11

**Table 2 entropy-22-00158-t002:** The various calculated correlation factors.

Correlation Factors	Direction of Adjacent Pixels		
	Horizontal	Vertical	Diagonal
Plain	0.9706	0.9841	0.9639
Cipher	0.0058	0.0033	0.0010
Plain	0.9195	0.9005	0.8696
Cipher	0.0013	0.0025	0.0010

**Table 3 entropy-22-00158-t003:** Average performance of NPCR (number of changing pixel rate) (%).

Images	Lena Image			Baboon Image		
	R	G	B	R	G	B
Proposed System	99.615	99.62	99.617	99.6140	99.6073	99.6292
[36]	99.60	99.61	99.61	99.6083	99.6065	99.6094
[37]	99.60	99.60	99.60	99.6138	99.6053	99.6182
[38]	99.6119	99.6097	99.6136	99.5864	99.5864	99.5864

**Table 4 entropy-22-00158-t004:** Average performance of UACI (unified averaged changed intensity) (%).

Images	Lena Image			Baboon Image		
	R	G	B	R	G	B
Proposed System	33.4732	33.3428	33.4647	33.4843	33.4690	33.4965
[36]	33.56	33.45	33.49	33.4939	33.4295	33.4856
[37]	33.369	33.43	33.37	33.4712	33.4265	33.4705
[38]	33.4811	33.4652	33.4907	33.4834	33.4639	33.2689

**Table 5 entropy-22-00158-t005:** Information entropy.

Images	Lena Image			Baboon Image		
	R	G	B	R	G	B
Proposed System	7.9973	7.9975	7.9975	7.9970	7.9978	7.9987
[36]	7.9971	7.9971	7.9971	7.9926	7.9926	7.9926
[37]	7.9819	7.9814	7.9829	7.9881	7.9871	7.9863
[38]	7.9973	7.9973	7.9973	7.9896	7.9896	7.9896

**Table 6 entropy-22-00158-t006:** Mean square error (MSE) between cipher images of Lena.

Images	$x_{1}$	$x_{2}$	$x_{3}$	$x_{4}$	MSE
Cipher $C_{1}$	$0.12+10^{-15}$	0.23	0.34	0.45	9.758
Cipher $C_{2}$	0.12	$0.23\times 10^{-15}$	0.34	0.45	9.614
Cipher $C_{3}$	0.12	0.23	$0.34\times 10^{-15}$	0.45	9.520
Cipher $C_{4}$	0.12	0.23	0.34	$0.45\times 10^{-15}$	9.6729

**Table 7 entropy-22-00158-t007:** MSE between cipher images of the baboon.

Images	$x_{1}$	$x_{2}$	$x_{3}$	$x_{4}$	MSE
Cipher $C_{1}$	$0.12\times 10^{-15}$	0.23	0.34	0.45	10.869
Cipher $C_{2}$	0.12	$0.23\times 10^{-15}$	0.34	0.45	10.725
Cipher $C_{3}$	0.12	0.23	$0.34\times 10^{-15}$	0.45	10.8432
Cipher $C_{4}$	0.12	0.23	0.34	$0.45\times 10^{-15}$	10.631

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
