# Peer review of "A Novel Color Image Encryption Algorithm Based on Hyperchaotic Maps and Mitochondrial DNA Sequences"

_entropy, 2020, doi:10.3390/e22020158_

Round 1

Reviewer 1 Report

Authors have reported a new encryption method. It is original and interesting. My comments and suggestions are given as.

(1) In sub-section 5.1, how to calculate the data ‘1.6777’ and ‘2.03451’? Some information are needed here.

(2) In sub-section 5.4, the values of parameters should be given for the decrypted images.

(3) In Fig.7 and Fig. 8, the original images can be deleted, because the images are shown in Fig.3.

(4) the following papers are related with this work.

[1] J. Wu, J. Shi, T. Li, A Novel Image Encryption Approach Based on a Hyperchaotic System, Pixel-Level Filtering with Variable Kernels, and DNA-Level Diffusion, Entropy 2020, 22: 5.

[2] Z. Liu, Q. Guo, L. Xu, M. A. Ahmad, S. Liu, Double image encryption by using iterative random binary encoding in gyrator domains, Optics Express, 2010, 18: 12033-12043.

Reviewer 2 Report

I would reject this paper based on below comments.

What is 'wellbeing' in the abstract?  'profits accomplished' does not make any sense. Please quantify 'outstanding encryption'. What is E curve. Please define.  ' length of keyspace must be very large', Please write a value. How much? ' brute-force violence'? The correct term is a brute-force attack. I think authors have tried to reduce the plagiarism and hence it resulted to change the terminology of image encryption.  Define the x-axis and y-axis of histogram tests. Entropy Effect? Please consider rewriting the draft paper. Please compare the proposed scheme with the latest schemes.   Future work is missing.

Reviewer 3 Report

Image encryption algorithm is presented.

The article has serious flaws.

Reviewer 4 Report

This paper shows a new chaotic encryption method and it well is decoded. Chaotic system is exploited to generate chaos sequence and authors use other robust techniques to improve the performance. The results show it obtains better performance than other algorithms. However, I have some questions below:

1. The abstract can be extended to explain the idea more evident than the current vision.

2. I still concern the description of the encryption steps, and it is not very clear. What is more, how do you to select the parameters?

3. Why your algorithm can be obtained better results than using other methods?

4. I strongly recommend authors to release the source code along with the submission, since the image processing based projects are typically open-source oriented to facilitate a fair assessment of the performance of the proposed methods for the community.

5. I would appreciate a deeper discussion on why the proposed method performs better than the others.

6. Some essential references are missing in the current version, e.g., Cross-utilizing hyperchaotic and DNA sequences for image encryption.

7. Last but not least, the author should pay attention to several grammar issues and inappropriate usages. The authors should carefully proofread this paper and correct all the typos in the revision.

Round 2

Reviewer 2 Report

Authors have revised the draft but there are still some issues.

There are still some typos for example line 51, ''encryption .In'' Line 51, ''offered an image''? I would recommend writing as; Askar ,A. A. Karawia et al. presented an image encryption system. Authors must not use such terminology. Technical writing must be improved before acceptance.  For the statement, ''As one of the recognized methods before
32 image transmitting is to convert it to an unintelligible form'', authors must provide references such as; https://doi.org/10.1007/s11042-015-2973-y; and https://doi.org/10.1007/s00521-016-2405-6  Please use a space between 'lamda3 and are'. See line 106. The statement ''Mitochondrial DNA is a variety of DNA situated external the core in the fluid portion of the cell 110 (cytoplasm) and internal cell organelles'' is sloppy, please rewrite. The statement ''DNA encoding has enormous data bulk so, it can be used in cryptography'' is correct? Please justify? Bulky data is used in encryption? I would strongly recommend to proof-read the draft paper. Check English grammar and sentence structure of ''The structure of the proposed cryptosystem demonstrated in Fig. 1, it involves two main phases.''.  In histogram plot x-axis is not correct. It should be pixel value of the green channel. What is a blue scale or green scale value? Please provide any reference for such terminology.  In conclusion '' in the sake of achieving''? Could you please use correct English.

Reviewer 3 Report

Authors have revised the first manuscript.

I am not satisfied with the author's reply.

In my opinion, the authors did not take a professional approach to conducting the experiments and designing the article.

Despite the iThenticate report, I still think that the already well-known research in Article https://doi.org/10.1117/1.JEI.26.1.013021 is largely repeated.

There are countless typos- rows 51, 209, 245 (CF) but 247 (CC),

I can not understand rows 203-207.

I don't see where are the real initial key values and wrong key values.

The authors have to compare the key size of the algorithm with other closely related algorithms.

I want to see values from the paper 10.1117/1.JEI.26.1.013021 with comparing with the proposed algorithm values.

My final decision is again Reject.

Reviewer 4 Report

My concerns are well addressed.

Round 3

Reviewer 3 Report

Thanks for answers.

This manuscript is a resubmission of an earlier submission. The following is a list of the peer review reports and author responses from that submission.